# Self-Assembled Triphenylphosphonium-Conjugated Dicyanostilbene Nanoparticles and Their Fluorescence Probes for Reactive Oxygen Species

**DOI:** 10.3390/nano8121034

**Published:** 2018-12-12

**Authors:** Wonjin Choi, Na Young Lim, Heekyoung Choi, Moo Lyong Seo, Junho Ahn, Jong Hwa Jung

**Affiliations:** 1Department of Chemistry and Research Institute of Natural Science, Gyeongsang National University, Jinju 52828, Korea; cwj1685@gnu.ac.kr (W.C.); skdud325@gnu.ac.kr (N.Y.L.); smile377@gnu.ac.kr (H.C.); 2Composites Research Division, Korea Institute of Materials Science, Changwon 51508, Korea

**Keywords:** dicyanostilbene, triphenylphosphonium, self-assembly, ROS detection

## Abstract

We report self-assembled novel triphenylphosphonium-conjugated dicyanostilbene-based as selective fluorescence turn-on probes for ^1^O_2_ and ClO^−^. Mono- or di-triphenylphosphonium-conjugated dicyanostilbene derivatives **1** and **2** formed spherical structures with diameters of ca. 27 and 56.5 nm, respectively, through π-π interaction between dicyanostilbene groups. Self-assembled **1** showed strong fluorescent emission upon the addition of ^1^O_2_ and ClO^−^ compared to other ROS (O_2_^−^, ^•^OH, NO, TBHP, H_2_O_2_, GSH), metal ions (K^+^, Na^+^), and amino acids (cysteine and histidine). Upon addition of ^1^O_2_ and ClO^−^, the spherical structure of **1** changed to a fiber structure (8-nm wide; 300-nm long). Upon addition of ^1^O_2_ and ClO^−^, the chemical structural conversion of **1** was determined by FAB-Mass, NMR, IR and Zeta potential analysis, and the strong emission of the self-assembled **1** was due to an aggregation-induced emission enhancement. This self-assembled material was the first for selective ROS as a fluorescence turn-on probe. Thus, a nanostructure change-derived turn-on sensing strategy for ^1^O_2_ or ClO^−^ may offer a new approach to developing methods for specific guest molecules in biological and environmental subjects.

## 1. Introduction

Generally, self-assembly of aggregation-induced emission (AIE)-active molecules in aqueous solution depends on intermolecular hydrogen bonding, π-π stacking, and hydrophobic interactions that form nanoscale architectures, such as spheres, rods, and fibers, with fluorescence enhancement [1,2]. Nanostructures formed via molecular self-assembly involving stimuli-responsive properties have great potential for biological and environmental applications [3,4,5,6]. Despite the utilization of dual effects on fluorescence turn-on originating from intramolecular electron transfer and intermolecular self-assembly [7], fluorogenic or chromogenic sensing of biologic or environmental species with selective turn-on detection have rarely been reported due to the difficulty associated with the design and synthesis of fluorescence probes. Strong interactions between acidic protons and solvent molecules cause act as a fluorescence quenching factor by photoelectron transfer (PET). Therefore, the self-assembled AIE fluorescence probes with hydrophobic positive charge are useful as sensing materials in biological field [8,9,10].

Studies related to optical detection using chromogenic or fluorogenic chemoprobes have been performed for biological and environmental applications of singlet oxygen (^1^O_2_) and hypochlorite (ClO^−^), such as the main component of a cleaning agent for industrial wastewater and cancer treatments that destroy tumor cells [11,12,13]. Fluorescent probes that demonstrate rapid response, high sensitivity, and technical simplicity are attractive tools for analyte monitoring [14], and fluorescent probes with a reaction-induced signal have been designed to detect a specific signal for singlet oxygen or hypochlorite [15]. In addition, a variety of fluorescent probes for ROS detection have been reported [11,16]. For example, a probe for singlet oxygen primarily produced detection signals by the addition reaction of singlet oxygen to anthracene or the rhodamine backbone [17,18]. In addition, an Ir (III) complex-linked coumarin 314 derivative has been used to identify a ratiometric signal through ^1^O_2_-mediated abstraction of the α-H from the tertiary amine [19]. In a probe for hypochlorite detection, fluorescent probe reactions with hypochlorite, such as oxidative cleavage reaction of double bonds (C=C, C=S, C=N, and N=N) or oxidative–hydrolysis reaction of amide, diphenyl ether, and thioether, produced detection signals from the reaction product [20].

In particular, depending on the self-assembly conditions, amphiphilic molecules with AIE have attracted significant attention due to the ease of controlling the fluorescence of an aggregate, which facilitates the construction of fluorescent turn off-on systems [21]. To obtain fluorescence turn-on signals by the reaction of a probe with an analyte, AIE molecules, such as tetraphenylethylene (TPE), are typically applied for ROS detection [22]. Dicyanodistyrylbenzenes are p-conjugated molecules with various optical properties, such as AIE in emission properties and tunable luminescence emission [23]. Compared to homologous a-cyanostilbenes, emission spectra occur at higher wavelengths due to their longer conjugation length [24]. Therefore, dicyanostilbene-linked amphiphilic molecules are a promising turn-on fluorescent sensor for singlet oxygen (^1^O_2_) and hypochlorite (ClO^−^). Thus, we designed triphenylphosphonium (TPP)-conjugated dicyanostilbene derivatives as amphiphilic molecules for turn-on detection of singlet oxygen (^1^O_2_) and hypochlorite (ClO^−^). TPP lipophilic cations were linked to dicyanostilbene to construct more rigid nanostructures in aqueous solution (i.e., the lipophilic cations enhance the stability of organic nanostructures by preventing solvent hydration and providing temperature or pH resistance) [25]. Here, we report the self-assembly properties of mono- or di-TPP-conjugated dicyanostilbene derivatives (**1** and **2**) and their behaviors as selective turn-on fluorescence probes toward singlet oxygen (^1^O_2_) and hypochlorite (Figure 1). In Scheme 1, we represented the detection strategies of ROS with turn-on fluorescence of self-assembled probe **1**. The self-assembled probe **1** with turn-off changed to turn-on through morphological transformation from sphere to continuously networked fibrous structures.

## 2. Materials and Methods

### 2.1. Reagents and Instruments

All reagents were purchased from Sigma-Aldrich (Yongin, South Korea). The solvent was purchased from Samchun Pure Chemicals (Pyeongtaek, South Korea) and used with further purification. ^1^H and ^13^C NMR spectra were measured using a Bruker DRX 300 spectrometer (Bruker). Furthermore, the mass spectra were measured using a JEOL JMS-700 mass spectrometer (JEOL Ltd., Tokyo, Japan). In addition, a Thermo Evolution 600 UV-vis spectrophotometer (Thermo Fisher Scientific, Waltham, MA, USA) was used to obtain the absorption spectra in the solution, and the fluorescence spectra were recorded using a RF-5301PC spectrophotometer (Shimadzu Corp., Kyoto, Japan).

### 2.2. Synthesis of Compound **3**

Compound **3** was synthesized using a previously reported method. Here 1,3-dibromopropane (2.5 mL, 24.57 mmol), *p*-hydroxybenzaldehyde (2.0 g, 16.38 mmol), and K_2_CO_3_ (3.39 g, 24.57 mmol) were dissolved in acetone (75 mL). The reaction mixture was heated to 50 °C for 12 h. Then, the reaction mixture was cooled to room temperature, filtered, and concentrated under reduced pressure. The crude product was purified using silica gel and eluted with ethyl acetate and hexane of compound **3**: ^1^H NMR (300 MHz, DMSO-*d*_6_) δ 9.88 (s, 1H), 7.90–7.86 (d, 2H), 7.17–7.14 (d, 2H), 4.23–4.19 (t, 2H), 3.70–3.66 (t, 2H), 2.32–2.24 (m, 2H); ^13^C NMR (75 MHz, DMSO-*d*_6_) δ 190.89, 163.89, 132.18, 130.38, 118.48, 114.99, 69.20, 65.91, 32.27, 29.78.

### 2.3. Synthesis of Compound **2**

Compound **3** (1.71 g, 7.03 mmol) and p-xylene dicyanide (0.5 g, 3.2 mmol) were dissolved in ethanol (30 mL). Sodium methoxide (0.35 g, 6.4 mmol) was added, and the solution was heated at reflux under N_2_ atmosphere. After 12, the reaction mixture was then cooled to room temperature, and the solid was filtered and washed with methanol: (yellow solid, 55% yield); ^1^H NMR (300 MHz, DMSO-*d*_6_), δ (ppm): 8.07 (s, 2H), 8.00–7.97 (d, 4H), 7.86 (s, 4 H), 7.17–7.14 (d, 4H), 4.22–4.18 (t, 4H), 3.70–3.68 (t, 4H), 2.34–2.25 (m, 4H); ^13^C NMR (75 MHz, DMSO-*d*_6_), δ (ppm): 160.3, 143.2, 134.4, 131.7, 126.0, 124.8, 118.5, 116.0, 105.1.

### 2.4. Synthesis of Probe **1**

Compound **2** (0.5 g, 0.82 mmol) was dissolved in acetonitrile (100 mL) with triphenylphosphine (1.3 g, 4.92 mmol). The mixture was heated to 85 °C for 48 h. Acetonitrile was removed under vacuum, and the precipitated yellow solid was collected by recrystallization. Then, the product was purified by flash chromatography using dichloromethane: (yellow solid, 61% yield); ^1^H NMR (300 MHz, DMSO-*d*_6_), δ (ppm): 8.10 (s, 2H), 8.00–7.76 (m, 42H), 7.14–7.12 (t, 4H), 4.25–4.18 (t, 4H), 3.81–3.71 (t, 4H), 2.02(s, 4H); ^13^C NMR (75 MHz, DMSO-*d*_6_), δ (ppm): 160.6, 143.4, 165.5, 134.8, 134.2, 134.1, 131.8, 130.9, 130.7, 127.0, 126.6, 119.4, 118.7, 118.2, 115.6, 106.9; FT-IR (cm^−1^): 2209, 1599, 1510, 1434, 1245, 1183, 1110; ESI-MS (m/z): 485.42 [**1** + H]^2+^, 1052.00 [**1** + Br]^+^.

### 2.5. Synthesis of Probe **2**

Compound **2** (0.5 g, 0.82 mmol) was dissolved in acetonitrile (100 mL) with triphenylphosphine (0.43 g, 1.64 mmol). The mixture was heated to 85 °C for 48 h. Acetonitrile was removed under vacuum, and the precipitated yellow solid was collected by recrystallization. Then, the product was purified by flash chromatography using dichloromethane/methanol (10:1). (yellow solid, 22.3% yield); ^1^H NMR (300 MHz, DMSO-*d*_6_), δ (ppm): 8.09 (s, 2H), 8.01–7.76 (m, 23H), 7.14–7.11 (s, 4H), 4.25–4.19 (s, 4H), 3.84–3.67 (s, 4H), 2.23–2.21 (s, 2H), 2.05 (s, 2H); ^13^C NMR (75 MHz, DMSO-*d*_6_), δ (ppm): 161.5, 143.9, 136.1, 135.9, 130.8, 130.1, 126.2, 119.2, 117.6, 107.7, 68.2, 33.8, 31.2; FT-IR (cm^−1^): 2214, 1594, 1508, 1434, 1508, 1434, 1259, 1176, 1109, 687; ESI-MS (m/z): 789.17 [**2** + H] ^+^.

### 2.6. Synthesis of **1-Ref**

**1-Ref** was synthesized by the reported method [S1].; ^1^H NMR (300 MHz, DMSO-*d*_6_), δ (ppm): 10.33 (s, 2H), 7.98 (s, 2H), 7.92–7.87 (d, 4H), 7.82 (s, 4H), 6.95 (s, 2H), 2); ^13^C NMR (75 MHz, DMSO-*d*_6_), δ (ppm): 160.4, 143.0, 133.4, 132.7, 126.5, 125.9, 117.4, 117.0, 105.2; FT-IR (cm^−1^): 3200, 2222, 1609, 1592, 1514, 1440, 1300, 1173.

### 2.7. Fluorescence Spectroscopy

A 1-cm long cuvette was used in the fluorescence assay. The sample was excited at 367 nm, and the emission was collected from 500 nm. ROS detection experiments were performed three times. To detect the ROS by probe **1**, 1 mL of standard solution (pH = 7.4) was first added to the cuvette. Then, the hydroxyl radical was generated via Fenton reaction with different amounts of Fe^2+^ and H_2_O_2_ (Fe^2+^/H_2_O_2_ = 1:10). After incubation with the probe for 15 min. For the selectivity experiment, hydroxyl radical (^•^OH) was generated via Fenton reaction (Fe^2+^/H_2_O_2_ = 200 μM, 2000 μM). Superoxide anion (O_2_^•−^) was derived from dissolved KO_2_ (200 μM) in the DMSO solution. Hypochlorite anion (ClO^−^) was provided by NaClO (200 μM). Nitric oxide (NO) and nitroxyl (HNO) were derived from a solution of S-nitroso-N-acetyl-DL-penicillamine and Angeli’s salt, respectively. ^1^O_2_ was generated by the reaction of H_2_O_2_ (200 μM) with NaClO (200 μM). Other species (10 equiv.) were prepared by dissolving in aqueous solutions at pH 7.4. All experiments were performed after incubation with the appropriate ROS/RNS for 10 min at room temperature.

### 2.8. Determination of Limit of Detection

The limit of detection (LOD) of probe **1** for ClO^−^ and ^1^O_2_ was determined as 33 μM and 56 μM, respectively. The LOD was calculated using the following equation, where σ is the standard deviation of the blank measurements and s is the slope of the calibration plot.
LOD = 3 × σ/s(1)


## 3. Results and Discussion

### 3.1. Characterization of Self-Assembled Probes **1** and **2**

Probes **1** and **2** were synthesized by a reaction of bromine-modified dicyanostilbene with TPP in acetonitrile following a previously reported method. Compounds **1** and **2** were confirmed by ^1^H and ^13^C NMR, ESI-MS, and FT-IR spectroscopy (Appendix A). ^1^H NMR data indicate that probes **1** and **2** demonstrated only a (Z)-form originating from the alkene peak at 8.04 in the dicyanostilbene moiety.

Since TPP-appended dicyanostilbene derivatives often demonstrate amphiphilic properties [26]. we observed the self-assembling behaviors in aqueous solution. In the aqueous solution (1% DMSO) at 25 °C, the UV-Vis absorption bands of **1** (25 μM) and **2** (6.25 μM) appeared at 360 nm and 371 nm, respectively. The UV-Vis absorption bands originated from π-π transitions of the dicyanostilbene moiety and shifted to a longer wavelength by increasing the temperatures of the aqueous solution (1% DMSO) (Figure 2). These red shifts were due to the formation of self-assembly via *J*-aggregation [27]. Note that the λ_max_ shift of probe **2** was smaller than that of probe **1**, which indicates that, compared to probe **2**, probe **1** formed more stable self-assembly in aqueous solution. The fluorescence spectra of probes **1** and **2** (excitation wavelength: **1** at 360 nm and **2** at 371 nm) were obtained by change of temperature (Appendix A). Weak fluorescence bands for probes **1** and **2** were observed at 512 nm due to the PET from fluorophore to TPP [28]. Due to the change from self-assembly to de-assembly, the fluorescence intensities decreased as the temperature increased. The fluorescence intensity of probe **1** decreased at 35–45 °C, whereas the fluorescence intensity of probe **2** decreased at 25–35 °C. We also measured the temperature-dependent ^1^H NMR spectra of probes **1** and **2** to obtain the key factor in the formation of self-assembly in DMSO-*d*_6_/D_2_O (99/1, *v*/*v*%) (Appendix A). The tendency of the chemical shift of an alkene peak in probe **1** was similar to that of probe **2**; however, the shift of probe **1** was greater than that of probe **2**. In addition, the interaction between alkene groups may affect enhancement of self-assembly stability. However, compared to probe **2**, the stabilization enhancement of self-assembled probe **1** led to intrinsic fluorescence quenching due to the reduced distance between dicyanostilbene and TPP via *J*-aggregation, as well as the effect of TPP groups inducing PET from dicyanostilbene.

We also observed morphologies of self-assembled **1** and **2** using atomic force microscopy (AFM) (Figure 2 and Appendix A). The AFM image of probe **1** showed a spherical structure with a diameter of ca. 22–32 nm (Figure 2b and Appendix A). The spherical nanoparticle should form by intermolecular dipole-dipole interaction and π-π stacking [29]. Similarly, probe **2** showed a spherical structure with a diameter of ca. 53–63 nm (Figure 2d and Appendix A). The size difference of self-assembled spheres **1** and **2** was due to the TPP group and the binding strength of the π-π interaction between the alkene groups, as shown by the ^1^H NMR data. Thus, we recognize that the size of the self-assembled spherical nanoparticles was determined by the strength of the intermolecular interactions. Based on UV-Vis, PL, and ^1^H NMR experiments, we conclude that **1** and **2** formed self-assembled spherical nanoparticles by *J*-aggregation with dipole-dipole interaction between the alkene groups in the dicyanostilbene moiety. The stability of self-assembled **1** was greater than that of self-assembled probe **2** (i.e., the TPP substitution effect accompanied fluorescence turn-off). We inserted a table involving summarized characteristics of the probes **1** and **2** (Appendix A).

### 3.2. ROS-Sensing Ability of Self-Assembled Probes **1** and **2** in Aqueous Solution

Self-assembled probes **1** and **2** with negligible fluorescence emission bands were used in aqueous solution to apply a turn-on fluorescence probe to highly ROS. The fluorescence spectral changes of self-assembled probe **1** were observed upon addition of several species related to ROS, such as singlet oxygen (10 equiv. of ^1^O_2_) and other ROS (10 equiv. of O_2_^−^, H_2_O_2_, NO, TBHP, ClO^−^, ^•^OH, GSH, cysteine, histidine, K^+^, and Na^+^) in water at pH 7.4 (Figure 3). Upon treatment with 10 equiv. of ^1^O_2_ and ClO^−^, a marked strong green emission at 520 nm was observed in under 5 min, indicating that ^1^O_2_ and ClO^−^ reacted with self-assembled **1** rapidly at room temperature. In addition, the fluorescence intensity of self-assembled **1** in the presence of ^1^O_2_ and ClO^−^ was enhanced by 2.3 and 2.7 times, respectively, compared to self-assembled **1** in the absence of ^1^O_2_ and ClO^−^. In contrast, significant selective changes in the emission were not observed upon addition of O_2_^−^, H_2_O_2_, NO, TBHP, ClO^−^, ^•^OH, GSH, cysteine, histidine, K^+^, and Na^+^ (Figure 3), indicating that these ROS species, amino acids, and metal ions did not react to **1**. The large difference in the fluorescence images between ^1^O_2_, ClO^−^, and other ROS was observed after treatment of **1** (Figure 3). Their fluorescence enhancement of self-assembled **1** in the presence of ^1^O_2_ or ClO^−^ indicated that the complex between **1** and ^1^O_2_ or ClO^−^ hinders PET [18]. The non-emission of self-assembled **1** upon addition of other oxide species, such as O_2_^−^ and NO, may be less reactive with 1 than that when ^1^O_2_ or ClO^−^ are added [30]. The sensing ability of self-assembled **2** was also evaluated under the same conditions. Self-assembled **2** exhibited fluorescence enhancement upon addition of O_2_^−^, ^•^OH, and ClO^−^. A hydrolysis reaction of the bromide group in self-assembled **2** with O_2_^−^ or ^•^OH did occur. The hydroxyl group in self-assembled **2** might induce fluorescence turn-off through the electron transfer mechanism [31]. Therefore, the fluorescence enhancement effect of self-assembled **2** was weaker than that of **1**. The slight turn-on effects of **2** in the presence of O_2_^−^, ^•^OH, and ClO^−^ may be caused by the reaction of a –Br group with ROS (O_2_^−^, ^•^OH, and ClO^−^), as indicated by the IR data (Appendix A).

To quantitatively investigate the reactivity and spectral changes of **1** (25 μM) upon addition of ^1^O_2_ and ClO^−^, fluorescence titrations were performed by adding ^1^O_2_ and ClO^−^ (0–375 μM) in water (containing 1% DMSO) at room temperature (Figure 4). The fluorescence intensity of **1** at 520 nm, which originates from dicyanostilbene moiety, was enhanced drastically during the titration process. In addition, at less than **2** equiv. of ^1^O_2_ and ClO^−^, an excellent nonlinear correction between fluorescence intensity and the concentration of ^1^O_2_ and ClO^−^ was obtained with *R*^2^ = 0.9923 for ^1^O_2_ and *R*^2^ = 0.9633 for ClO^−^, indicating that the ratio of fluorescence intensity at 520 nm was enhanced as a nonlinear function of ^1^O_2_ and ClO^−^ concentration. This turn-on mechanism can be attributed to the cooperative effects of AIEE of dicyanostilbene and blocking of the photoinduced electron transfer process. The detection limits of self-assembled **1** for ClO^−^ and ^1^O_2_ were 33 μM and 56 μM, respectively (Appendix A) [S2].

To further evaluate the utility of **1** as a selective fluorescence probe for ^1^O_2_ and ClO^−^, the competition-based fluorescence emission changes of self-assembled **1** upon addition of various biologically relevant species and ROS (i.e., ClO^−^, ^1^O_2_, O_2_^−^, H_2_O_2_, NO, TBHP, ClO^−^, ^•^OH, GSH, cysteine, histidine, K^+^, and Na^+^) were investigated in aqueous solution (Appendix A). In binary system, probe **1** showed a strong green emission with ^1^O_2_ or ClO^−^ except for cysteine, histidine and NO. The emission change with cysteine was due to that the phenoxy oxygen of **1** may be interacted with -SH group in cysteine [32,33]. In addition, histidine molecules were due to their bioactive properties inducing the reaction of histidine with ^1^O_2_ or ClO^−^ prior to probe **1** [33]. Low turn-on emission of probe **1** in presence of K^+^ and Na^+^ ions can be caused by the charge interaction of cation with ^1^O_2_ or ClO^−^ prior to react with probe **1**. As expected, the fluorescence intensities of self-assembled **1** in the presence of ^1^O_2_ and ClO^−^ were unchanged by treatment of other species, such as O_2_^−^, H_2_O_2_, NO, TBHP, ^•^OH, GSH, K^+^, and Na^+^, indicating that self-assembled **1** is a new selective turn-on fluorescence probe of ^1^O_2_ and ClO^−^ in a mixture of other species.

### 3.3. ROS-Mediated Fluorescence Turn-On Mechanism of Self-Assembled Probe **1** in Aqueous Solution

The mechanism of the reaction between self-assembled **1** and ^1^O_2_ or ClO^−^ was further studied. First, the bonding cleavage of **1** upon addition of ^1^O_2_ or ClO^−^ was confirmed at the molecular level. The mechanism of the reaction between self-assembled **1** and ^1^O_2_ or ClO^−^ was further studied. First, the bonding cleavage of **1** upon addition of ^1^O_2_ or ClO^−^ was confirmed at the molecular level. The shape of the UV-Vis spectra of self-assembled probe **1** treated with ^1^O_2_ or ClO^−^ was almost same to that without ^1^O_2_ or ClO^−^ (Appendix A). On the other hand, the molecular structure of dicyanostilbene was conserved when ^1^O_2_ or ClO^−^ was added to self-assembled **1**. We observed FAB-Mass and IR spectral changes of **1** with ^1^O_2_ or ClO^−^ (Appendix A). After treatment with ^1^O_2_ or ClO^−^, a mass value of 667.3 was obtained. This value indicates that the oxygen atom adjacent at dicyanostilbene reacted with ^1^O_2_ or ClO^−^ and then formed a phenol moiety (Appendix A). Furthermore, the IR spectra of **1** upon addition of ^1^O_2_ or ClO^−^ showed new peaks at 900–1100 cm^−1^ (OH bending vibration), and peaks in the range of 3100–3700 cm^−1^ were widened due to the formation of the phenolic OH group in **1** (Appendix A). In addition, by comparing the IR data (1000–2400 cm^−1^) of self-assembled **1** with or without analyte, we confirmed that the structure of the dicyanostilbene moiety was conserved while self-assembled **1** reacted to ^1^O_2_ or ClO^−^. A dicyanostilbene derivative possessing OH group (**1-Ref**) was synthesized to assign OH groups on IR and NMR data (Appendix A). In Appendix A, the OH groups the product obtained from probe **1** in the presence of ^1^O_2_ or ClO^−^ correspond with the OH peaks compound **1-Ref**. However, the product obtained from probe **1** with ^1^O_2_ or ClO^−^ was produced one −OH as shown in Appendix A, but not two −OH groups. We measured zeta potential of probe **1** and probe **1** treated with ^1^O_2_ or ClO^−^ to further confirm about its structure change (Appendix A). The zeta potential of the probe **1** was determined to be 32.46 mV; however, the zeta potential of the probe **1** treated with ^1^O_2_ or ClO^−^ was 21.43 mV and 23.16 mV respectively; this was due to the formation of OH group. These results strongly indicated that the OH group originated from the reaction of probe **1** with ^1^O_2_ or ClO^−^ had been successfully formed.

Furthermore, the morphological change of self-assembled **1** was observed by treatment with ^1^O_2_ or ClO^−^ by AFM. After treatment with ^1^O_2_ or ClO^−^, the spherical nanoparticle formed via self-assembly of **1** changed to a fiber structure (Figure 5 and Appendix A).

Therefore, we propose a schematic illustration of the reaction-based morphological change of self-assembled **1** during the reaction that progressed upon adding ^1^O_2_ or ClO^−^ (Figure 5). To the best of our knowledge, the selectivity of probe **1** for ^1^O_2_ could be described by two-step reactions. First, an aliphatic carbon adjacent to the oxygen atom was attacked by ^1^O_2_ [34]. Then, water reacted with the ^1^O_2_-derived activated group to form a hydroxyl end group in the dicyanostilbene moiety. Due to the structural stabilization effect of the fully-conjugated dicyanostilbene moiety, alkene groups in dicyanostilbene may be less reactive than the carbon atoms adjacent to the oxygen atom of probe **1**. 

## 4. Conclusions

We have synthesized novel triphenylphosphonium-strapped dicyanostilbene derivatives (probes **1** and **2**) and characterized their self-assembly properties by using spectroscopic analysis. Owing to the TPP substitution effects, probe **1** were formed more stable self-assembly than that of probe **2**. The spherical structure of self-assembled probe **1** were shown selectively turn-on fluorescence upon addition of ^1^O_2_ or ClO^−^. Moreover, the morphology of self-assembled probe **1** was changed from sphere to fibrous structures in the presence of ^1^O_2_ or ClO^−^. We also proved that the generation of OH group-substituted TPP-dicyanostilbene caused by the reaction between oxygen atoms adjacent dicyanostilbene of probe **1** and ^1^O_2_ or ClO^−^. Thus, we could propose continuous fibrous-mediated AIEE effect-based turn-on sensing mechanism of probe **1** for ^1^O_2_ or ClO^−^ by using molecular and nanometer level analysis. Based on our results, it is expected that a nanostructure change-derived turn-on sensing strategy for ^1^O_2_ or ClO^−^ may offer a new approach to develop methods in biological and environmental subjects.

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
