# Peer review of "Self-Assembled Triphenylphosphonium-Conjugated Dicyanostilbene Nanoparticles and Their Fluorescence Probes for Reactive Oxygen Species"

_nanomaterials, 2018, doi:10.3390/nano8121034_

Round 1
Reviewer 1 Report
Authors addressed constructions and properties of self-assembled nano-particles as selective fluorescence turn-on probes for 1O2 and ClO-, using triphenylphosphonium-conjugated dicyanostilbene. Author carried out suitable works for the development of turn-on typed selective fluorophore, and this paper is logically described.
Accordingly, I recommend this paper to publish nanomaterials after a revision related to following comments.
1. Authors should summarize the fundamental characteristic of the fluorophores such as absorption and fluorescence maxima, and further quantum yield, using an adequate table.
2. Authors described those turn-on behavior taking place by chemical reaction between the fluorophores and 1O2 or ClO-, suggesting that the corresponding group transformed from triphenylphosphonium group to OH group. The characteristic using NMR and IR is sufficient, but, if possible, authors should separately prepare the compound possessing OH group and should confirm and compare the structures and the self-assemble behavior.
Author Response
Response to Reviewer 1 Comments
We appreciate your favorable comments and thank you for your recommendation of our manuscript (nanomaterials-403745) for publication in nanomaterials after minor revision. In addition, we have revised our manuscript according to your suggestions.
1. For “Authors should summarize the fundamental characteristic of the fluorophores such as absorption and fluorescence maxima, and further quantum yield, using an adequate table”:
Reply: As you suggested, the characteristics of absorption and fluorescence was summarized in a Table S1. We added a Table S1 in the revised manuscript.
2. For “Authors described those turn-on behavior taking place by chemical reaction between the fluorophores and 1O2 or ClO-, suggesting that the corresponding group transformed from triphenylphosphonium group to OH group. The characteristic using NMR and IR is sufficient, but, if possible, authors should separately prepare the compound possessing OH group and should confirm and compare the structures and the self-assemble behavior”:
Reply: The product obtained from the probe 1 after treatment with 1O2 was separated by filtration. Then, we measured 1H NMR spectrum of the product. We also synthesized the compound possessing OH group as a reference to compare with reaction product between probe 1 and 1O2. IR and NMR spectra of the product obtained from 1 after treatment with 1O2 were newly inserted in the revised manuscript (Figure S7A and S8). We also added the explanation in the revised manuscript as follows;
“A dicyanostilbene derivative possessing OH group (1-Ref) was synthesized to assign OH groups on IR and NMR data (Figure S7A and S8). In Figure S7 and S8, the OH groups the product obtained from probe 1 in the presence of 1O2 or ClO- correspond with the OH peaks compound 1-Ref. However, the product obtained from probe 1 with 1O2 or ClO- was produced one –OH as shown in Figure S8, but not two –OH groups.
Thank you once again for your kind suggestions and generosity with your time.
Reviewer 2 Report
This is an excellent contribution on self-assembly of chromophore molecules with function of detection of reactive oxygen species. It satisfies professional requirements on this research subject. Therefore i recommend publication of this work in Nanomaterials.
1) Please provide couterion information in chemical formulae.
2) References are well cited. However, addition of very recent reviews on functional self assembly would make impression much better.
(i) Self-assembly of bodipy-derived extended pi-systems, Bull. Chem. Soc. Jpn. 2018, 91, 100-120.
(ii) Self-assembly of discrete organic nanotubes, Bull. Chem. Soc. Jpn. 2018, 91, 623-668.
(iii) Self-assembly as a key player for materials nanoarchitectonics, Sci. Technol. Adv. Mater., in press. DOI: 10.1080/14686996.2018.1553108
3) For the AFM image, please provide color column for height information (not only cross-sectional profile).
4) Please describe more about future possibilities of biological practical applications with concrete examples. This short description would make this work much more attractive in a wide range of research fields.
Author Response
Response to Reviewer 2 CommentsWe appreciate your favorable comments and thank you for your recommendation of our manuscript (nanomaterials-403745) for publication in nanomaterials after minor revision. In addition, we have revised our manuscript according to your suggestions.
1. For “Please provide counter ion information in chemical formulae”:
Reply: Thank you for careful comments, we corrected the chemical formulae (Figure 1) in the revised manuscript.
2. For “References are well cited. However, addition of very recent reviews on functional self-assembly would make impression much better”:
Reply: As you commented, we added new references in the revised manuscript; i) Self-assembly of bodipy-derived extended pi-systems, Bull. Chem. Soc. Jpn. 2018, 91, 100-120; ii) Self-assembly of discrete organic nanotubes, Bull. Chem. Soc. Jpn. 2018, 91, 623-668; iii) Self-assembly as a key player for materials nanoarchitectonics, Sci. Technol. Adv. Mater., in press. DOI: 10.1080/14686996.2018.1553108
3. For “For the AFM image, please provide color column for height information (not only cross-sectional profile)”:
Reply: We added the AFM image involving color column in Figures 2 and S9 in revised manuscript.
4. For “Please describe more about future possibilities of biological practical applications with concrete examples. This short description would make this work much more attractive in a wide range of research fields”:
Reply: We added a concluding remark in the revised manuscript as follows:
“We have synthesized novel triphenylphosphonium-strapped dicyanostilbene derivatives (probe 1 and 2) and characterized their self-assembly properties by using spectroscopic analysis. Owing to the TPP substitution effects, probe 1 were formed more stable self-assembly than that of probe 2. The spherical structure of self-assembled probe 1 were shown selectively turn-on fluorescence upon addition of 1O2 or ClO-. Moreover, the morphology of self-assembled probe 1 was changed from sphere to fibrous structures in the presence of 1O2 or ClO-. We also proved that the generation of OH group-substituted TPP-dicyanostilbene caused by the reaction between oxygen atoms adjacent dicyanostilbene of probe 1 and 1O2 or ClO-. Thus, we could propose continuous fibrous-mediated AIEE effect-based turn-on sensing mechanism of probe 1 for 1O2 or ClO- by using molecular and nanometer level analysis. Based on our results, it is expected that a nanostructure change-derived turn-on sensing strategy for 1O2 or ClO- may offer a new approach to develop methods in biological and environmental subjects.
Thank you once again for your kind suggestions and generosity with your time.
Reviewer 3 Report
The manuscript by Choi et al. describes two self assembling structures for 1O2 and ClO-.
Detection principle is based on fluorescence emission enhancement upon partial degradation of the self assembling structure.
There are numerous fluorescence based probes for detection of ROS in the literature, and the work could probably be more appropriate for an analytical journal. That being said, the idea of using self assembled nanomaterials is intriguing and worth investigation. I have a couple of issues that prevent me from suggesting publication, but if the authors can address these issues I would favorably see the manuscript published.
1) it is not completely clear to me why the self assembly stability NMR studies are conducted in almost pure DMSO while all other tests are performed in water.
2) Fluorescence dependence on pH of probe 1 seems quite remarkable. I see that a 40% increase is observed depending on pH. While this is not a severe issue in designing an analytical probe, the authors should probably remove their "pH stability" claim.
3) Detection limits reported in page 7, row 16 are astonishingly high: could the authors check whether there is some typo in calculating them?
4) Discussion on the mechanism of disassembly should be removed unless there are strong evidences for the degradation mechanism of this particular probe (i.e. the authors should remove the "ester like" part). Additionally, it would be great if the authors could quantify, e.g. by MRM mass spectrometry, the amount of degraded product identified in figure 5c. This would corroborate the claimed mechanism.
5) did the author try to further confirm their claim of disassembly as a driving force for fluorescence enhancement by means of light scattering measurement? changes in count number would surely confirm their findings.
If the authors can address these issues, I think that the manuscript would be an excellent contribution to Nanomaterials.
Author Response
Response to Reviewer 3 CommentsWe appreciate your favorable comments and thank you for your recommendation of our manuscript (nanomaterials-403745) for publication in nanomaterials after major revision. In addition, we have revised our manuscript according to your suggestions.
1. For “It is not completely clear to me why the self assembly stability NMR studies are conducted in almost pure DMSO while all other tests are performed in water.”:
Reply: Since compound 1 was formed a strong self-assembly in 100% of D2O, 1H NMR spectrum of compound 1 was extremely broadened in 100% of D2O. Thus, we measured 1H NMR spectrum of 1 to obtain information for a driving force of self-assembled 1 in a mixed DMSO and D2O (1:1 v/v). We added this explanation in the revised manuscript.
2. For “Fluorescence dependence on pH of probe 1 seems quite remarkable. I see that a 40% increase is observed depending on pH. While this is not a severe issue in designing an analytical probe, the authors should probably remove their "pH stability" claim.”:
Reply: According to a reviewer comments, we deleted a result about "pH stability" in the revised manuscript.
3. For “Detection limits reported in page 7, row 16 are astonishingly high: could the authors check whether there is some typo in calculating them?”:
Reply: “Yes”. Detection limit of probe 1 in this manuscript was calculated by reported method previously in the literatures (Anal. Chem. 2017, 89, 7210−7215. Anal. Chem., 1983, 55, 712. RSC Adv., 2016, 6, 53912). We cited the references (Ref. S2) in the revised manuscript. As you mentioned above, we found calculation error for LOD from calibration curve. So we inserted correct LOD values in revised manuscript.
4. For “Discussion on the mechanism of disassembly should be removed unless there are strong evidences for the degradation mechanism of this particular probe (i.e. the authors should remove the "ester like" part). Additionally, it would be great if the authors could quantify, e.g. by MRM mass spectrometry, the amount of degraded product identified in figure 5c. This would corroborate the claimed mechanism.”:
Reply: According to the reviewer’s comments, the description of degradation mechanism was revised in the manuscripts as follows.
“To the best of our knowledge, the selectivity of probe 1 for 1O2 could be described by two-step reactions. First, an aliphatic carbon adjacent to the oxygen atom was attacked by 1O2 [35]. Then, water reacted with the 1O2-derived activated group to form a hydroxyl end group in the dicyanostilbene moiety. Due to the structural stabilization effect of the fully-conjugated dicyanostilbene moiety, alkene groups in dicyanostilbene may be less reactive than the carbon atoms adjacent to the oxygen atom of probe 1.”
As you suggested, we separated the product obtained from the probe 1 after treatment with 1O2 was separated by filtration. Then, we measured 1H NMR spectrum of the product. We also synthesized the compound possessing OH group as a reference to compare with reaction product between probe 1 and 1O2. IR and NMR spectra of the product obtained from 1 after treatment with 1O2 were newly inserted in the revised manuscript (Figure S7A and S8). We also added the explanation in the revised manuscript as follows;
“A dicyanostilbene derivative possessing OH group (1-Ref) was synthesized to assign OH groups on IR and NMR data (Figure S7A and S8). In Figure S7 and S8, the OH groups the product obtained from probe 1 in the presence of 1O2 or ClO- correspond with the OH peaks compound 1-Ref. However, the product obtained from probe 1 with 1O2 or ClO- was produced one –OH as shown in Figure S8, but not two –OH groups.
5. For “Did the author try to further confirm their claim of disassembly as a driving force for fluorescence enhancement by means of light scattering measurement? changes in count number would surely confirm their findings.”:
Reply: It is well known that DLS data of fiber structure materials are not able to have reliability. Therefore, we added a new result involving zeta potential measurement of probe 1 with 1O2 or ClO- in the revised manuscript. And additional descriptions were added as follow;
“We measured zeta potential of probe 1 and probe 1 treated with 1O2 or ClO- to further confirm about its structure change (Figure S10). The zeta potential of the probe 1 was determined to be 32.46 mV; however, the zeta potential of the probe 1 treated with 1O2 or ClO- was 21.43 mV and 23.16 mV respectively; this was due to the formation of OH group. These results strongly indicated that the OH group originated from the reaction of probe 1 with 1O2 or ClO- had been successfully formed.”
Thank you once again for your kind suggestions and generosity with your time.